# Digital Habits of Users in the Post-Pandemic Context: A Study on the Transition of Mexican Internet and Media Users from the Monterrey Metropolitan Area

Daniel Javier de la Garza Montemayor [1,*], Daniel Barredo Ibáñez [2,3] and Mayra Elizabeth Brosig Rodríguez [4]

1 Departamento de Administración, Universidad de Monterrey, San Pedro Garza García 66238, Mexico
2 Department of Journalism, Emergia Fellow, Universidad de Málaga, 29010 Málaga, Spain
3 Fudan Development Institute, Fudan University, Shanghai 200437, China
4 Facultad de Contaduría Pública y Administración, Universidad Autónoma de Nuevo León, San Nicolás de los Garza 66455, Mexico
* Correspondence: daniel.delagarza@udem.edu

**Abstract:** The COVID-19 pandemic accelerated the transformation of digital interactions, a development that has been growing in the last decade. Digital platforms have become indispensable in the institutional communication of public and private organizations. The magnitude of this change was evident during the pandemic at a time when several countries implemented social distancing measures to contain the contagion. This situation caused a certain degree of user dependence on information and communication technologies. The objective of this research is to analyze the time of use, the changes, and habits of digital consumption at the beginning and during the COVID-19 pandemic in the metropolitan area of Monterrey, Nuevo León, Mexico. Access to both social networks and digital entertainment platforms was examined during the period in which Internet users had resumed a large part of face-to-face activities, and 1500 questionnaires were conducted considering the current data of the population of the urban area according to what was reported by the INEGI (National Institute of Statistics and Geography). The results indicate that after COVID-19, a digital transformation was accelerated, and, in that period, social media helped to build trust according to the users consulted in the Monterrey metropolitan area. However, trust was given at an interpersonal level due to motivations such as the prior user relationships in offline spaces and not so much because of the institutional campaigns that were behind the digital transformation.

**Keywords:** internet; COVID-19; Mexico; internet; social networks

## 1. Introduction

During the COVID-19 pandemic, there was an undeniable acceleration in digital interaction. This was largely motivated by the restrictions imposed by different governments to prevent the virus from proliferating [1].

It is possible to argue that throughout this time, some aspects, both positive and negative, can be observed in the daily life of people as a direct consequence of the increase in digital media usage. Throughout a cycle that has lasted about three years, people have used technology to hold work meetings, continue their studies, stay both informed and entertained, and above all, to keep in touch [2,3].

In the second year of the pandemic, two of the authors of this research conducted a study of university students in the important city of Monterrey, Mexico (and its metropolitan area). One of the constructs not only best explained the increase in technologies in terms of interpersonal communication but also helped to understand their experience was that of trust, and the hypothesis of this research is that there was a general increase in the use of digital information media [4].

To increase the representativeness of the sample, a second survey was carried out in 2022, but in this case, it was aimed at the general population of the same geographical area. Before that, some aspects of the previous study were corrected, and a pilot test was carried out that was statistically validated. In the following pages, and before we present the methodology section, we are going to reflect on the relation between trust and digital platforms, consumption of social networks, and digital entertainment in the era of COVID-19.

The objective of this research is to analyze the time of use, the changes, and habits of digital consumption at the beginning of and during the COVID-19 pandemic in the metropolitan area of Monterrey, Nuevo León, Mexico. The specific objectives are to describe the consumption and time of use of the information media, compare the increase in the use of the media at the beginning and during the COVID-19 pandemic, and describe user trust in the media and the time spent on entertainment content. The novelty of our study is based, first, in the shortage of other works which have examined the digital habits in the specific moment of COVID-19's first year, but secondly, we can argue this is a relevant work which contributes to the research line of digital consumption and trust in a way that it has been proven the singularity of the interpersonal communication use of social media during the analyzed period and with the chosen sample. We also aimed to analyze what behaviors due to digital consumption habits have increased during the pandemic and the use of streaming platforms. This topic has been chosen due to the lack of enough empirical studies in the area or due to the studies being conducted in other countries during the pandemic.

## 2. Trust and Digital Platforms

Since the consolidation of digital tools, the trust factor has been analyzed as a fundamental variable in the consolidation of new technologies. Trust is a central element that allows digital media to be an ideal space to receive information from users but also helps to generate credibility in service providers [5].

Since the rise of social networking sites, the connection between trust and the use of information and communication technologies has been analyzed repeatedly. There was a time when the association between the use of technology and the involvement of young people in matters of public interest was positive [6].

It is important to understand how trust can become a predictor variable of consumption by users. For example, there is empirical evidence that familiarity with and therefore trust in certain Internet sites can help explain online purchase intentions [7,8].

Moreover, evidence suggests that the elements that can help build trust among users is that they interact with brands and institutions with which they are familiar. In other words, prior knowledge is required to allow Internet users to trust certain online advertising campaigns, regardless of whether they are aware of how digital marketing works [9].

Other variables that can influence this trust-building relationship between social media and Internet users are both real social interactions and perceived economic benefits. These factors can boost electronic commerce at a point in time even in a competitive environment [10].

In building trust, it is essential both for public institutions that seek to communicate institutional messages to the public and for those private companies whose purpose is to gain the trust of customers. If the public feels free to interact on digital sites, there is a greater propensity to get messages from brands and institutions [11,12].

In the context of COVID-19, thanks to the prevailing connection, the dissemination of information on public health issues was possible in a way that would have been unthinkable in the past, but there is evidence that these tools were not used responsibly by everyone. For this reason, it is possible to argue that political, business, academic, and opinion leaders have an important responsibility so that these can be channeled in the most ethical manner [13].

This last idea has been confirmed in other investigations. Social media can contribute to building both trust or mistrust among the public, and this was especially true during the pandemic. The traditional communication model has been transformed in recent years, and

relevant actors can contribute to a large extent to communication that can constructively influence public debate [14].

It is also important that the quality of information disseminated through digital platforms has the capacity to be understood by different segments of the population. The disinformation that proliferates in cyberspace is generally clear to the average user, so truthful information must be accessible as well [15].

The pandemic was essential to increase the visibility of some of the problems that would inevitably arise in the information age. According to an investigation carried out in the first year of the pandemic in Italy, social media became a fundamental space to spread vital information, but these platforms also were responsible of publishing some false news that altered the public opinion. In the second case, a series of government institutions intervened to combat disinformation that was out of control [16].

For this reason, it can be affirmed that the responsibility of generating reliable information that contributes to creating a constructive environment depends on a multiplicity of actors. Prestigious institutions such as universities, health centers and participatory members of society can also influence this in a positive manner. All of them had an important role in combating the misinformation that proliferated during the most critical stage of the pandemic, where there was a lack of accurate information about the virus in the early stages [17].

There is another factor that can influence the extent to which the tools contribute to generating a relationship of trust between the users, which are the sources of information themselves. If the person who shares or disseminates any specific news within the community is a known actor, such as a family member, trust in the information received can increase significantly, whether it is valid or not [18].

This idea was confirmed with another investigation in which the conclusions pointed out that although there are no clear parameters that make it possible to clarify whether people grant credibility to a certain source, it can be greatly influenced if the person disseminating the information is a specialist on the matter. In other words, the medium outlet does not guarantee that the information is reliable, and rather it is about the trajectory, personal and professional prestige of a certain account that gives the message legitimacy [19].

Misinformation during the COVID-19 pandemic greatly influenced the effectiveness of governments and the health sector in countering the deadly virus. In a study carried out in Lebanon, it was shown that trust in the news spread by both moral and formal authorities had contributed significantly to spreading myths and rumors about the virus on social media [20].

The power of digital platforms was clearly demonstrated during the pandemic, but these outlets are far from being the only means of communication. When it is necessary to position a message for large groups of the population, it is convenient to adopt different sources of information. Traditional media and digital media are not necessarily mutually exclusive [21].

The effects of digital media can become a public health issue. During the pandemic, large groups of the population had to be immersed in digital platforms, transcending the so-called digital natives who had predominated with their presence in the previous years. There is evidence that older adults had anxiety derived from the information found in social networks, something that contributed to diminishing trust at a social level [22].

## 3. Consumption of Social Networks in the Era of COVID-19

During the pandemic, rumors were present in various media, but above all in virtual social networks. For this reason, it can be argued that it is important to know the effects of social media consumption on personal behavior since it has become an essential factor that can help describe behavior in times of crisis [23].

Social media was instrumental, as stated above, in conveying news about the pandemic, even though apocryphal news were also generated in the process. These media represent an opportunity for authorities because its influence can be positive if well managed [24].

It is also possible to argue that in more than one region it was confirmed that social networks were the most used means of communication in the new decade that began in 2020. What seemed like an emerging trend in recent years ended up crystallizing in the middle of a global emergency. During the strictest stage of the lockdown, people were able to express their emotions through these means, something that could have had a positive psychological effect [25].

This approach is shared by a study conducted in India during the first 2 years of the pandemic. That study concluded that active users of social networks have increased substantially since 2020. It warns about some of the risks posed by these tools, but concludes that their benefits are greater [26,27].

In that sense, it is relevant to point out that the exponential increase in the use of social networks was not restricted only to young people. Both minors and adults significantly increased their use of social networks during this time. One of the factors that significantly influenced the intensity of use was their need to find information in a time of crisis [28].

Similarly, there is evidence that seems to contradict the previous studies that have been mentioned. Some people had some degree of anxiety and presented a behavior that could be categorized as addictive in terms of social media consumption. In other words, the consumption of digital media can be an emotional and mental health issue that should be taken into consideration in further studies [29,30].

One way to counteract the negative side effects of the exposure of information disseminated through social networks can cause is through education. As the use of digital platforms has increased significantly, recommendations have also emerged to strengthen digital literacy as an antidote to misinformation and the potential discomfort caused by this situation [31,32].

On the other hand, it is essential to distinguish that not all social networks have the same effects on users. An investigation in Canada confirmed that it was the social network Twitter on which there was a greater dissemination of fake news, but it is also true that there was a differentiation between digital media and the traditional media: while those who consumed the former more showed they had less solid information about the coronavirus, those that consumed the latter had notions that were closer to reality [33].

During the pandemic, the rise of the TikTok social network was notable. In general, the overall use of virtual social networks grew during this period, as was mentioned before, but the TikTok platform experienced a sustainable growth that allowed new possibilities for the dissemination of information, both for entertainment and informational purposes [34,35].

## 4. Digital Entertainment Consumption

One of the industries that grew the most, at least in terms of content consumption, was the digital entertainment industry. In the long confinement that was experienced in several countries, digital platforms were a refuge for millions of people in the world [36].

Since 2020, there have been notable changes in entertainment consumption. Although it was expected from the first year that the patterns would change once the restrictions that accompanied the first stage of the pandemic were lifted, it was clear that some trends would remain in a post-covid world [37].

Immediately, the film industry faced significant challenges as ratings dropped dramatically. Several important festivals and premieres had to be postponed [38].

Changes in habits occurred in groups of all ages. In other words, the adoption of technology as a means of entertainment saw a significant increase in both youth and adults, including those who were middle-aged [39].

During this season, it became apparent that both the entertainment industry in general and the home theater market had changed dramatically. We are moving from a model in which cable service prevailed as a means of obtaining additional entertainment options to a time when different streaming platforms are competing for the market [40].

The crisis caused by the pandemic had an impact on the entire entertainment industry, notably in the US case. The studios had to improvise some strategies, such as the premiere

of some productions in a hybrid format. Evidence from a qualitative study suggested that audience habits would hardly be the same after this moment [41].

It is possible to argue that the changes in the industry were evident before the pandemic; for example, they pose a huge challenge to the business model. Consumers look for variation in content, and that is why platforms such as Netflix began offering original content in 2013 [42,43].

However, high production costs and competition create an uncertain future for the streaming industry. Although the use of streaming platforms increased significantly during the pandemic, this also represented a huge financial challenge for the main companies that offered original content [44,45].

However, it is also true that the entertainment industry is undergoing a transformation process in which other factors beyond consumption habits have changed. The offer in terms of production was also apparent because consumers had access to a diversity of content, both local and international, which expanded the traditional offer that had prevailed in these media [46–48].

During this period, various studies were carried out that allowed us to better understand some of the factors that motivate consumers. We can highlight, among these factors, familiarity, social norms, as well as the perception of enjoyment. It was also found during this period that the time spent by users had increased in such a way that it interfered with other activities [49,50].

## 5. Methodology

To carry out this research, the data was collected using the questionnaire-type instrument named "Digital Consumption Habits", which obtained a Cronbach's Alpha of 0.925. The sample was of the stratified type, composed of 1552 participants, and the percentage of men was 46.2% and the percentage of women was 53.8%, while in the age range, 73.3% of the participants were between 18 and 22 years old, followed by the range between 22 and 33 years of age with 15.8%, and 10.5% of the participants were over 34 years old, proportionally distributed by the population size of the municipalities of the metropolitan area of Monterrey, Nuevo León, in Mexico. To carry out the proposed objectives, descriptive statistics techniques, percentage analysis, and student's T test for related samples were performed, and the analyses were performed using the SPSS Version 25 software. Some of the questions that the poll included are based on previous studies, conducted before and after the pandemic. The following dimensions were taken into consideration: consumption of information media, use of social networks, reliability of media, and consumption of digital platforms for entertainment purposes [51–55].

*Justification*

Monterrey, Nuevo León is an atypical case study within the Mexican Republic, as in the year 2020, 78.8% of the population had access to the internet in some way. At the same time, Monterrey is key to understanding the Mexican case. The capital of the State of Nuevo León has been in the third place in Mexico in terms of information and communication technologies [56].

## 6. Results

The COVID-19 virus pandemic transformed various aspects of society, including digital consumption habits [1–3]. To check to what extent those habits were transformed in Monterrey, the research took a sample of 1552 participants, of whom 46.2% were women and 53.8% were men. Figure 1 shows the distribution of the sample participants, where the municipality of Monterrey represents the first site with 27.1%, while the municipality of Apodaca represents the second site with 17.8%, and in the third site was the municipality of Guadalupe with 12.5% of the sample.

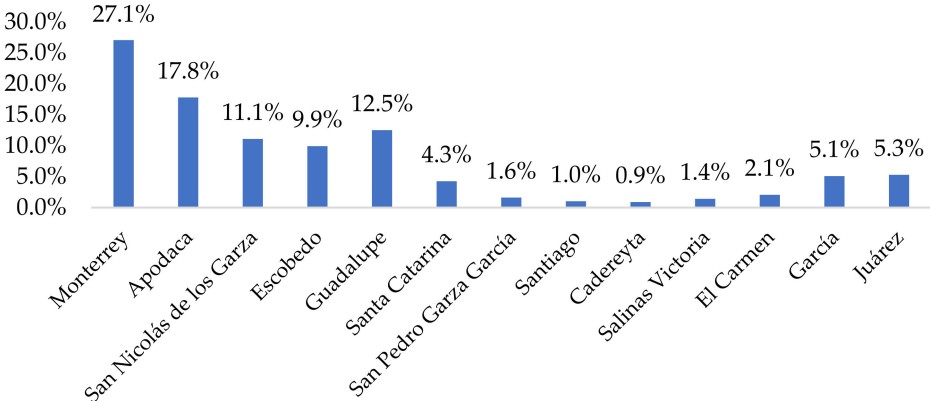

**Figure 1.** Percentages of participants according to the municipality that belongs to the metropolitan area of Monterrey.

To deeply understand the data, Figure 2 shows us the ranked distribution from highest to lowest percentage of "totally agree" for the information media which users said they consume at least once a day. WhatsApp has the highest response, with a value of 62.1%, followed by the Internet, with a value of 52.1%, and in third place is Facebook, with a value of 52%. The use of WhatsApp, the Internet sites, and Facebook occupy the first places for the consumption of information, and newspapers are located last. In other words, digital platforms are more frequently consumed than traditional media, which occupy the seventh position in this survey.

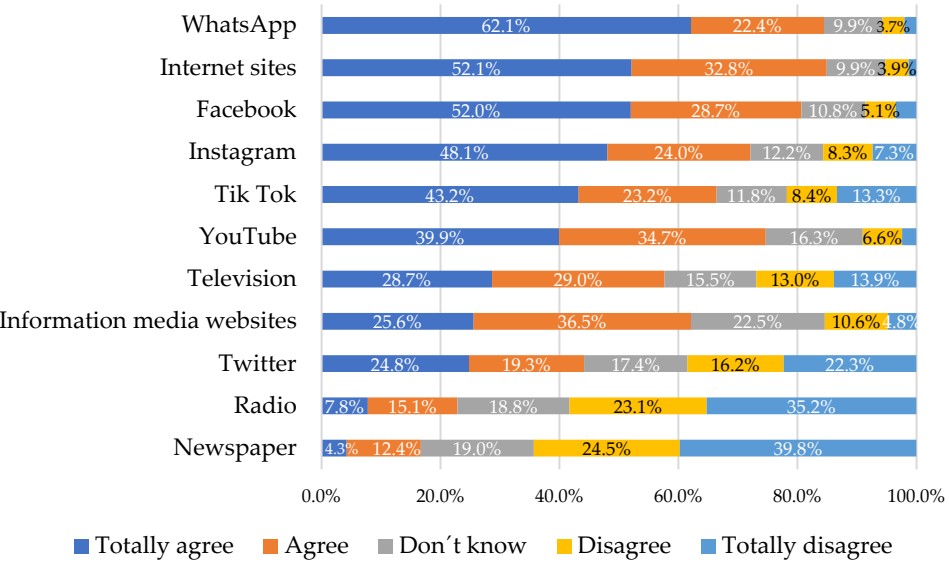

**Figure 2.** Distribution of percentages of responses to the consumption of information media.

Table 1 shows us the time that users spend on social networks, with the interval between 3 and 4 h occupying the first site with a value of 42.7%; in addition, the use dedicated by the users is also presented.

Table 2 shows us the result of the t-test. To compare the increase in consumption from at least 1 to 2 times per day, it can be seen that, in media such as television, websites, Facebook, Instagram, and Twitter, the $p < 0.05$, and therefore, they presented changes in the habit of consumption, while radio and YouTube presented changes in the interval signs, and it was also observed that $p > 0.05$ for TikTok, and WhatsApp media $p > 0.05$, so it can be concluded that there were no substantial changes in use, but it is to be noted that TikTok appeared during the pandemic, as is explained by [35].

**Table 1.** Percentages of time intervals of use in social networks and the Internet.

| Use Time | Approximately How Many Hours a Day Do You Dedicate to the Different Contents of Social Networks? | Approximately How Many Hours a Day Do You Spend Browsing the Internet? |
|---|---|---|
| 0 h | 1.7 | 3.5 |
| Between 1 and 2 h | 28.5 | 48.3 |
| Between 3 and 4 h | 42.7 | 27.5 |
| More than 5 h | 27.1 | 20.7 |

**Table 2.** "Student t" test for related samples in the media to analyze their increase at the beginning of and during COVID-19.

| | Paired Differences | | | | | T | GL | Significance (Bilateral) |
|---|---|---|---|---|---|---|---|---|
| | Average | Dev. Standard | Average Error | 95% Confidence Interval of the Difference | | | | |
| | | | | Inferior | Superior | | | |
| Television | −0.583 | 1.328 | 0.034 | −0.649 | −0.517 | −17.292 | 1551 | 0.000 |
| News media websites | −0.416 | 1.189 | 0.030 | −0.475 | −0.356 | −13.769 | 1551 | 0.000 |
| Internet | −0.275 | 1.012 | 0.026 | −0.326 | −0.225 | −10.713 | 1551 | 0.000 |
| Radio | −0.051 | 1.077 | 0.027 | −0.105 | 0.003 | −1.862 | 1551 | 0.063 |
| Facebook | −0.082 | 0.976 | 0.025 | −0.130 | −0.033 | −3.302 | 1551 | 0.001 |
| YouTube | −0.008 | 1.106 | 0.028 | −0.063 | 0.047 | −0.275 | 1551 | 0.783 |
| Instagram | 0.115 | 1.078 | 0.027 | 0.062 | 0.169 | 4.215 | 1551 | 0.000 |
| TikTok | 0.006 | 1.117 | 0.028 | −0.050 | 0.061 | 0.205 | 1551 | 0.838 |
| Twitter | 0.205 | 1.149 | 0.029 | 0.148 | 0.262 | 7.027 | 1551 | 0.000 |
| WhatsApp | 0.032 | 1.077 | 0.027 | −0.022 | 0.085 | 1.155 | 1551 | 0.248 |

Figure 3 shows us the distribution from highest to lowest of the response of "totally agree" for the trustworthiness of users towards the media, with television as the medium that users have as a higher percentage, with a value of 21.6%, followed by news media websites, with a value of 20.6%, and in third place is the written press, with a value of 16.4%. Television and newspapers continue to be reliable media as news sources. However, the newspaper, as observed in Figure 2, has a very low consumption.

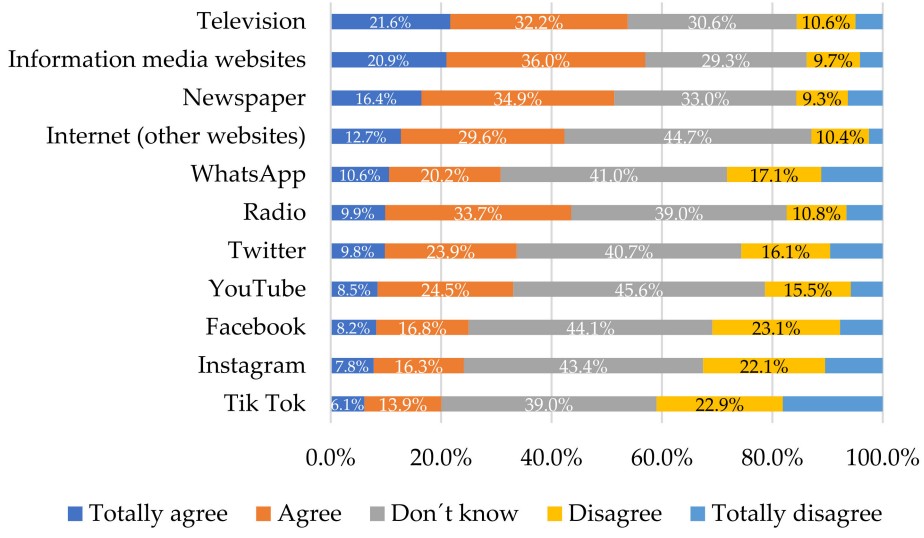

**Figure 3.** Distribution of percentages of responses to reliability in the media.

Table 3 shows the time that users spend per day on entertainment content. It is observed that the interval between 1 and 2 h is the one with the highest percentage, with

a value of 47.2%, followed by the interval between 3 and 4 h, with a value of 35.2%. A majority accepts that they spend time on entertainment content online daily.

**Table 3.** Time spent on different entertainment content.

| Use Time | Approximately How Many Hours a Day Do You Spend on Different Entertainment Content? |
|---|---|
| 0 h | 3.2 |
| Between 1 and 2 h | 47.2 |
| Between 3 and 4 h | 35.2 |
| More than 5 h | 14.4 |

Figure 4 shows us the distribution from highest to lowest of the percentages of the response "totally agree" on the situations that have increased during the COVID-19 pandemic, and it can be seen that the first site is for the use of social networks, which presents a percentage to the response of a value of 61%, followed by the use of text messages, with a response value of 58.2%, and in third place is the use of voice, with a response rate of 50.7%. The increase in social media use during the COVID-19 pandemic is clear, as is the use of texting and voiced notes, and emails are in last place.

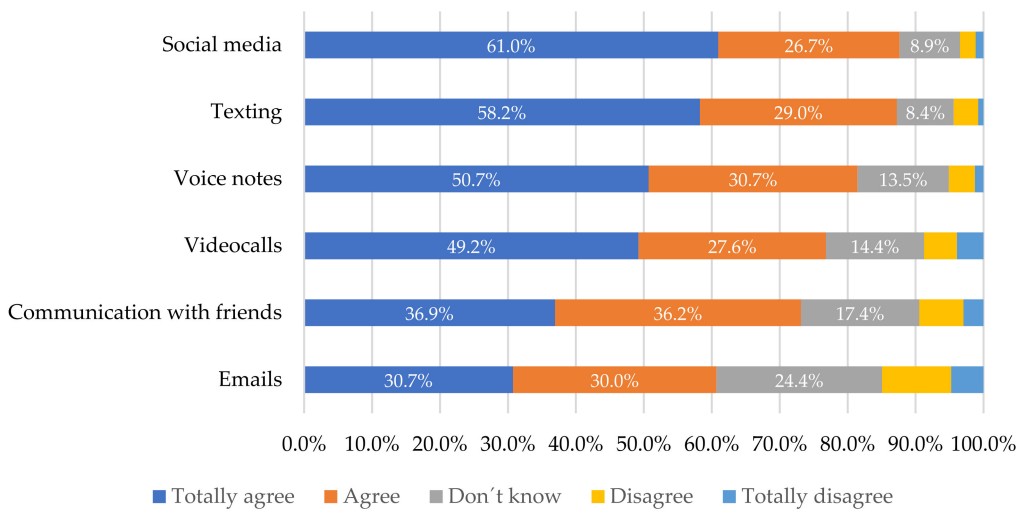

**Figure 4.** Distribution of percentages of responses to situations that have increased due to the COVID-19 pandemic.

The comparative analysis of the use of the media at the beginning and during the COVID-19 pandemic showed that for television media, websites, Facebook, Instagram, and Twitter, there were significant changes, while radio, YouTube, TikTok, and WhatsApp did not present significant increases in their use.

In Figure 5, we can see the percentages from highest to lowest of the responses "totally agree" for the situations of consumption of entertainment content, and it can be seen that the first place is occupied by the pleasure of watching digital entertainment platforms in times of rest, with a percentage of 47.2%, in second place is the relaxation from watching entertainment content, with a value of 40.5%, and in third place it is found that the contents have served for users to have topics of conversation, with a value of 39.8%. An important segment of those that responded to the survey used digital platforms not only for entertainment purposes, but they were also important in their interactions with others.

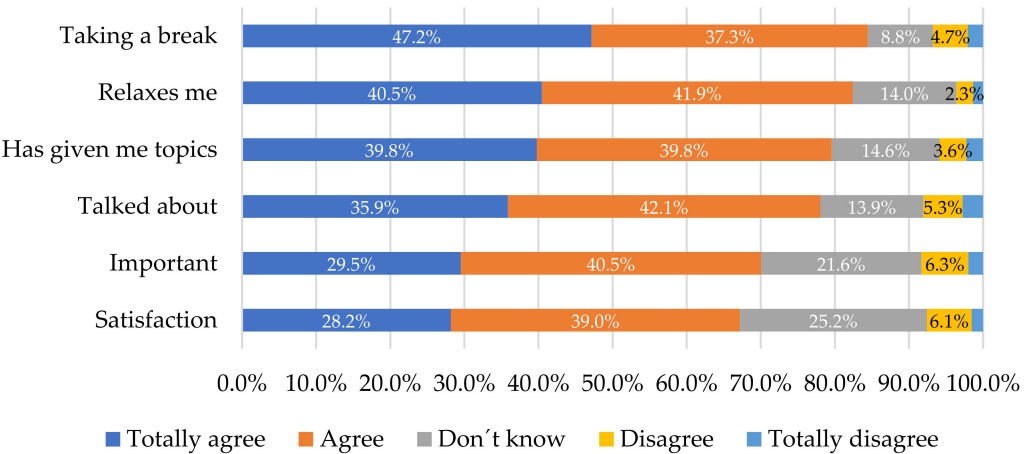

**Figure 5.** Distribution of percentages of responses to situations of consumption of entertainment content.

As we have observed, we can conclude that the consumption of certain information media had a significant increase in its use, such as television and other digital platforms such as Facebook and Instagram, and it was also observed that the consumption of entertainment content occurs in times of rest.

## 7. Discussion and Conclusions

COVID-19 was a factor that accelerated digital transformation thanks to the institutional campaigns [1] but also with the migration of the daily use both from professional and personal angles [2,3]. Digital habits, therefore, increased massively after the pandemic as a result of evident phenomena such as preventing isolation around the world. The novelty of this study, however, is associated concretely to the explanation of trust in Monterrey, one of the most important Mexican capitals.

Trust is a key to gain credibility [5] and, in the case of commercial organizations, it can drive higher sales [7,8,10]. In general, trust allows a deeper interaction with users [11,12] in a moment that traditional communication models have been eroded [14]. Moreover, trust is an essential vector that fosters social cohesion and social participation as it can also be described as a preventive factor to combat misinformation [18,19].

In this context of accelerated digital transformation and the need to stimulate trust among users, social media play a controversial role. As these platforms are widely accessed by users, they are an instrument for institutional communication [24], especially in a world crisis such as the COVID-19 pandemic. Social media are associated with a more diverse content, both locally and internationally [46–48], but these are controversial outlets as some studies confirm a link between consumption of social media and psychologically negative effects [29,30].

In our study, we have confirmed that, in the examined period, social media were used to build trust among users, but this occurred above all on an interpersonal level despite the mentioned institutional campaigns that were behind the digital transformation. According to the literature, that phenomena could be explained is as follows. When the scale of those media that have the greatest trust among citizens is considered, two conventional media types stand out, which are the press and television. It is also important to indicate that trust grows when connected to other people we know, as was also previously stated by other authors [18,19,21,33].

Those findings that ensure that there was a significant increase in the use of digital platforms during the pandemic are also confirmed. In this study we also seek to highlight which were those digital media that experienced the greatest growth during the pandemic, and we confirmed that they were mostly social networks. Others remained unchanged, but in no case was it confirmed that there was a decrease in these [26–28].

Even though digital media consumption prevailed over traditional media, in this study there is evidence that proves a coexistence of both channels as sources of information during the COVID-19 pandemic. In the changing environment [20,21] imposed by the generalized restrictions, trust can be a key to understand that the mentioned consumption coexists both in traditional and digital media. In a moment of collective uncertainty, users needed to check multiple points of view. That was, probably, a mechanism to prevent disinformation by contrasting the facts from a variety of sources. At the same time, as the lockdown restricted outdoor activities, there was more time available to obtain information and entertainment from mass media.

As it was highlighted in the results, this study confirms some trends that emerged in the context of the pandemic, such as the rise of the TikTok social network, which has played an important role in recent years [35,37] It is important to remember that this study was representative of the population of the metropolitan area of Monterrey, so it was not limited to a specific group of the population, like several studies that are based on technology. The sample also considered adults of all ages. There are coincidences with studies that show that digital media was accessible to people with different characteristics, both in terms of occupation and age [15,22].

During and after the lockdown, many people continued to express themselves through electronic media and mostly avoided personal contact. This implied an adaptation process in interpersonal communication that had immediate effects on information consumption but could also anticipate other changes in social behavior [23,25].

In the case of digital platforms, the study also coincides with those investigations that stated that Internet users consume content influenced by their social environment, but also for the purpose of entertainment. Like most of the studies that were consulted, those who answered the survey confirmed that during the pandemic the use of digital platforms increased significantly [36,49,50].

It is important to establish that the study has certain limitations. In the first instance, when the instrument was applied, people in the city once again had a life without as many restrictions as the COVID-19 imposed. It is possible that the passage of time can help generate a more complete perception. On the other hand, the results are representative of an important state of a relevant Latin American country, but it is possible that the results are divergent in other areas of the region where there is a smaller or larger digital divide.

**Author Contributions:** Conceptualization, D.J.d.l.G.M.; methodology, D.J.d.l.G.M. and D.B.I.; software, M.E.B.R.; validation, D.J.d.l.G.M., D.B.I. and M.E.B.R.; formal analysis, D.J.d.l.G.M. and M.E.B.R.; investigation, D.J.d.l.G.M., D.B.I. and M.E.B.R.; resources, D.J.d.l.G.M., D.B.I. and M.E.B.R.; data curation, M.E.B.R.; writing—review and editing, D.J.d.l.G.M., D.B.I. and M.E.B.R.; visualization, D.J.d.l.G.M., D.B.I. and M.E.B.R.; supervision, D.J.d.l.G.M., D.B.I. and M.E.B.R.; project administration, D.J.d.l.G.M. All authors have read and agreed to the published version of the manuscript.

**Funding:** This research received no external funding.

**Informed Consent Statement:** Informed consent was obtained from all subjects involved in the study.

**Data Availability Statement:** Data on SPSS is available upon request.

**Conflicts of Interest:** The authors declare no conflict of interest.

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
