# Peer review of "Digital Habits of Users in the Post-Pandemic Context: A Study on the Transition of Mexican Internet and Media Users from the Monterrey Metropolitan Area"

_societies, doi:10.3390/soc13030072_

Round 1

Reviewer 1 Report

Dear authors!

I had the opportunity to read your article, which I find generally interesting.

Let me leave a few comments that I hope will make you take another look and improve the paper:

1. In the Abstract, I would like to see the result of the study

 2. The structure of the article

a. The objective of the study should be removed from the methodology

b. The hypothesis is not clear.

 3. The objective of the study.

·         The objective of the study should follow from the analysis of sources on the topic. Analyzing sources you should identify a gap in knowledge

·         The objective of this research (Lines 206-212) «to analyze the time of use, the changes, and habits of digital consumption at the beginning and during the COVID-19 pandemic» that you have set seems quite general. Has any research been done on this topic? What exactly is in these studies?

·         Also, the purpose of the research should be commensurate with the title of the article.

·         At the same time, the goals stated in the Abstract and in the article itself differ.

 4.Methodology

Lines 28-39 relate to the methodology and it is better to transfer them there, but also ....

• It is not clear how many respondents were in the first part of the study (students)

• How many respondents were interviewed in the second general part of the study

 5. Review of sources needs to be reworked.

Your analysis of the sources is interesting and important, but it is absolutely not clear from it why you took this particular topic. Were there articles and works devoted to your region, your country

 6. Work more carefully on references to sources

For example, you write, (lines 46-49) that

There was a time when the association between the use of technology and the involvement of young people in matters of public interest was positive, but there is evidence that things have changed since then.

At the same time, when talking about new changes, you refer to the article of 2009.

Author Response

We are thankful for all the observations that have been made in this review. The suggestions have improved our text. We changed the abstract, and the information that was missing has been included. We also improved the methodology section, according to the suggestions that were made. We have made clear why this subject was chosen, and also we took into consideration the last suggestion regarding the text from 2009. Also, the discussion and conclusions have been changed, so the justification of this article is stronger in this version.

Reviewer 2 Report

The article "Digital habits of users in the post-pandemic context. A study on the transition of Mexican Internet users from the Monterrey Metropolitan Area" addresses an interesting topic, namely because it considers the possibility of changing digital habits in the post-pandemic period. Despite the topic being interesting and the results presented could reveal the reality of a particular region, the article has several problems, both in terms of structure and content. The literature review is relevant, and considers some of the work carried out in recent years on media consumption habits, but also other important issues such as the dissemination of misinformation on social networks. Despite the literature review's relevance, several aspects could be deepened, namely considering the dependence on digital. However, it is mainly in the empirical part that the work presents the most significant problems. Although the methodological procedures are well explained, the presentation of the results is limited to describing what the authors found in the graphs, with no kind of discussion or crossing with the studies indicated in the literature review.

On the other hand, there are several problems in relation to the questions posed in the questionnaire, as it mixes, in the same questions, devices with applications, but also with platforms and the Internet itself. Several aspects also show a lack of verification of the text, for example, in line 234, where the values indicated in the text do not correspond to those presented in the graph. On line 296, it is unclear why the graphic appears in the middle of the text. Between lines 289 and 295, data that was previously presented is repeated. The same happens again between lines 305 and 307, where the same information already present in lines 275-278 is repeated. Graph 5 does not display any data. In addition to these aspects, and considering there is no discussion of the data, we verify conclusions only repeat what had already been said. Therefore there is no new information that results from this article. For this reason, we believe the article is not yet ready to be published and must be worked on so that it can be submitted again in the future.

Author Response

We are thankful for the very useful suggestions the reviewer has made, which contributes to improve our article. All the suggestions have been taken into consideration. First, we agree that further studies should deal with dependence on digital media. The discussion of the results has been improved, so they do cross with the literature that has been presented. About the lack of verification of the text, we can say that this has been revised and mistakes have been corrected. We believe the study has greatly improved, thanks to the suggestions that were made by this evaluation.

Reviewer 3 Report

The article is well-written and contributes to the field of social sciences. The information provided is adequate to determine movements in the consummation of entertainment platforms by Mexican users. Authors are invited to write the main result reaching in the abstract. As well as in the methodology or results sections, specify the age range and gender of the people surveyed to determine the influence of these two variables on the results, perhaps as future lines of research.

Author Response

We are thankful for the suggestions. The abstract now includes the main results that the research had.

Round 2

Reviewer 1 Report

Dear authors. The article has been significantly revised, but the following points have not been worked out:

1. The results of the study that you indicate in the abstract are too general and there is nothing new in them.

2. The purpose of the study.

• The purpose of the study should follow from the analysis of sources on the topic. Analyzing sources you should identify a gap in knowledge

3. Review of sources needs to be reworked.

Your analysis of the sources is interesting and important, but it is absolutely not clear from it why you took this particular topic. Were there articles and works devoted to your region, your country

Author Response

Dear reviewer,

Once again, we are grateful for your observations. We did address the three keys factors you pointed out (results, purpose, and sources). There are not enough sources in the region about this subject, as we do have them in other international studies that were made in the same period.

In the abstract, we gave more insight into the objectives and the results are better explained than before (Lines 10-12, 16-20). We also added an argument about the novelty of our research in the introduction, and why this subject was chosen (lines 51-60). Accordingly, our discussions have been improved (lines 364-371). To consolidate the explanation and importance of our results and how they are relevant to the overall study, new comments have been included in results section. We also improved the presentation of the graphs. Once again, we feel this article has been improved with your observations.

Sincerely,

The authors

Reviewer 2 Report

The article underwent a slight improvement, but the review did not consider several of the indicated aspects. Among the aspects that were not considered we highlight the presentation of results, where the author(s) limit their work to a description of the information found in the graphs, with no interpretation of the data. The way in which graphs and tables are introduced, as well as their quality, in the case of graphs, and the font, in the case of tables, do not seem to be entirely adequate either. The titles of the graphs do not appear in the correct place, as can be seen, for example, in line 325, relative to graph 4. The author(s) must follow the journal's rules and ensure that the document complies with the rules. The changes already made have improved the text, but flaws still need to be corrected.

Author Response

Dear reviewer,

We are thankful once again for your observations. We have taken all into consideration and made changes to our article. First, the presentation of the results now includes further comments which try to show the relevance of them, and how they relate to the objectives of the research. We also improved the presentation of the graphs. The titles and description of the information contained in the graphs are in the correct position. We also follow what the template of the journal indicates. We also improved the abstract, which now contains more insight into the objectives and results. The novelty of the research is also better justified now in the introduction. The discussion has also been improved, to provide more depth on the results and what we argue is their contribution. Once again, we feel your comments have been very useful.

Sincerely,

The authors